# Sedation Quality and Cardiorespiratory, Echocardiographic, Radiographic and Electrocardiographic Effects of Intramuscular Alfaxalone and Butorphanol in Spanish Greyhound Dogs

**DOI:** 10.3390/ani13182937

**Published:** 2023-09-16

**Authors:** Julio Fernández Castañer, Setefilla Quirós Carmona, Carmen Martínez Bernal, Juan Morgaz Rodríguez, Rocío Navarrete Calvo, María del Mar Granados Machuca

**Affiliations:** 1Veterinary Teaching Hospital, School of Veterinary Medicine, University of Córdoba, 14014 Córdoba, Spain; v72fercj@uco.es (J.F.C.); cmb676@yahoo.es (C.M.B.); 2Department of Animal Medicine and Surgery, School of Veterinary Medicine, University of Córdoba, 14014 Córdoba, Spain; v92moroj@uco.es (J.M.R.); nacar6@hotmail.com (R.N.C.); pv2grmam@uco.es (M.d.M.G.M.)

**Keywords:** alfaxalone, butorphanol, dogs, cardiorespiratory parameters, intramuscular, sedation, echocardiographic, radiographic, electrocardiographic

## Abstract

**Simple Summary:**

Sedation is sometimes necessary in veterinary practice to perform different medical procedures, such as echocardiography or radiography, due to uncooperative or aggressive patients. In patients with cardiovascular disease, anxiety needs to be reduced; however, hemodynamic conditions have to be preserved, so it is advisable to avoid sedatives such as alpha-2 agonists or phenothiazine, due to their cardiovascular side effects. Many studies support the cardiovascular stability obtained by the administration of alfaxalone, alone or in combination, in dogs. Butorphanol is a synthetic opioid characterized by minimal cardiovascular effects, making it an ideal drug for sedation protocols. We aimed to evaluate the hemodynamic impact of the administration of intramuscular alfaxalone and butorphanol in healthy dogs. The evaluation involved assessing the degree of sedation and the effects of the sedation on cardiorespiratory, echocardiographic, radiographic and electrocardiographic parameters at baseline and 30 min after the administration of sedation. Although statistically significant changes were observed in some of the studied parameters, no clinical alterations were found 30 min after drug administration. The results showed that alfaxalone and butorphanol at the doses used in this study provided moderate sedation for performing diagnostic procedures, and they caused mild hemodynamic and respiratory changes.

**Abstract:**

The quality of sedation and changes in cardiorespiratory variables after the intramuscular administration of alfaxalone and butorphanol in Spanish greyhound dogs were evaluated. Twenty-one adult dogs were included. The dogs received alfaxalone (2 mg/kg) and butorphanol (0.2 mg/kg) intramuscularly. Sedation scoring, cardiorespiratory parameters (including blood gas analysis), echocardiography, thoracic radiography and electrocardiography were performed before sedation and 30 min after drug administration. Moderate sedation was observed, and side effects, such as tremors, nystagmus and auditory hyperesthesia, were noticed. Statistically significant changes in heart rate, invasive blood pressure, pH, arterial saturation of O_2_ and partial pressure of O_2_ and CO_2_ were found. Echocardiographic variables, including end-diastolic volume, left ventricular diameter in diastole, aortic and pulmonic flow, diastolic transmitral flow and left atrial/aortic ratio, and electrocardiography parameters, including PQ interval and QT interval, showed statistically significant changes. In conclusion, the intramuscular administration of alfaxalone and butorphanol to healthy dogs produced moderate sedation with mild cardiorespiratory, echocardiographic and electrocardiographic changes, without alterations in cardiac size on radiographic images.

## 1. Introduction

Sedation is used in daily clinical practice to perform different minor medical procedures, to conduct diagnostic imaging studies and to manage aggressive, nervous or excited patients. In patients with cardiovascular disease, the preservation of the hemodynamic condition is important for diagnostic procedures that aim to investigate cardiovascular function. Moreover, it is recommended to avoid stress and excessive manipulation in these patients, so intramuscular sedation is useful when the patients cannot be handled awake to perform the diagnostic procedure. The sedative protocol used for this purpose should ideally reduce anxiety, provide immobilization and produce minimal or no cardiovascular changes [1]. In daily clinical practice, the use of alpha-2 agonist with butorphanol is routinely used for sedation in dogs, but the side effects of this combination have been described in different complementary tests, such as echocardiography [2,3] or evaluation of tear production [4]. A combination of midazolam and butorphanol has been used to produce sedation in dogs, due to its minimal cardiovascular effects, but this combination may induce excitation in healthy dogs [5]. Acepromazine, alone or in combination with opioid analgesics [6], has been studied in dogs. Acepromazine alone caused a decrease in systemic arterial blood pressure [1], and its combination with pethidine produced a decrease in the left ventricular internal dimensions in healthy greyhounds [7].

Alfaxalone is a synthetic neuroactive steroid that produces anesthesia through an interaction with the gamma aminobutyric acid (GABA)A receptor within the central nervous system [8], and its anesthetic effect has been published in different species [9,10,11,12,13,14]. In dogs, alfaxalone has been demonstrated to have a high margin of safety, producing minimal cardiorespiratory side effects and producing few or no cardiovascular changes when it is used at clinical doses [15]. These hemodynamic changes are dose-dependent, with supraclinical doses causing increased heart rate, hypotension and hypoventilation [16]. The intramuscular (IM) administration of alfaxalone has been reported alone [17] and in combination with other drugs in different species [18,19,20,21]. The combination of α-2-adrenergic drugs and alfaxalone is associated with substantial cardiorespiratory effects after IM administration [18,22]. Nevertheless, one study [23] showed a good quality of sedation with limited echocardiographic changes after the administration of 4 mg/kg of alfaxalone intramuscularly in adult dogs; however, side effects such as paddling, trembling and nystagmus were described.

Butorphanol is a kappa receptor agonist and a mu receptor antagonist that causes dose-dependent sedation with minimal cardiopulmonary depression [24]. Several studies describe the effects of intramuscular administration of butorphanol and alfaxalone with sedatives [18,21,22,25]. The IM combination of alfaxalone with opioids, such as methadone [26] or butorphanol [18], together with midazolam, has been studied. As a consequence of the addition of midazolam, an increase in behavioral side effects and variability in the quality of sedation were observed in both studies. However, the evaluation of the effects of the combination of alfaxalone and butorphanol in dogs is limited [25], and, to the author’s knowledge, the effects of the combination of these two drugs on echocardiographic, radiographic and electrocardiographic parameters in the dog have not been described.

The objective of this study was to evaluate the sedation quality and the effects on cardiorespiratory, echocardiographic, electrocardiographic and radiographic parameters of the intramuscular administration of alfaxalone in combination with butorphanol in healthy Spanish greyhound dogs. Our hypothesis is that the intramuscular combination of these drugs will provide a good quality of sedation and minimal variations in the studied parameters.

## 2. Materials and Methods

This observational, prospective and clinical study was performed at the Veterinary Teaching Hospital, University of Córdoba. It was approved by the Ethical Committee for Animal Welfare at the Teaching Hospital of the University of Córdoba (CEBAHCV 47/2020). All procedures were conducted in compliance with the ethical principles of good practice in animal experimentation and with the previous informed consent of the owners. 

### 2.1. Animals

Twenty-one client-owned, adult, sexually intact dogs (11 males and 10 females, mean body weight 23.1 ± 3.8 Kg) scheduled to undergo neutering at the Veterinary Teaching Hospital of the University of Cordoba were enrolled in the study. All dogs were greyhounds (Spanish greyhound), and the inclusion criteria required the dogs to be aged >1 year and be in good health, according to their history and the results of a complete physical examination, preoperative serum biochemistry and hematologic analyses. Dogs with evidence of cardiopulmonary dysfunction on clinical, electrocardiographic or echocardiographic examination and animals currently on medication were excluded.

### 2.2. Study Design

Animals were admitted two days before surgery. On the morning of the study, a 20-gauge cannula (VasoVet^®^, B Braun GmbH, Melsungen, Germany) was placed into the pedal artery 20 min after the application of topical anesthesia (EMLA 5% cream). Prior to taking the baseline (BL) measurements, all dogs were allowed to acclimate to the room. The dogs were placed in right lateral recumbency. Their heart rate (HR), respiratory rate (RR), temperature (Tº) and invasive arterial blood pressure (IBP) were recorded from the pedal artery. An arterial blood sample was taken, and blood gases were measured. A thoracic right lateral view radiograph was taken, and a six-lead ECG and standard echocardiography were performed.

Twenty minutes after baseline measurements were taken, each dog was sedated via the administration of 0.2 mg/kg of butorphanol (Torbugesic^®^ 10 mg/mL Zoetis, Barcelona, Spain) and 2 mg/kg of alfaxalone (Alfaxan 10 mg/mL; Vetoquinol, Spain), intramuscularly in the hind limb. Dogs were kept in a quiet room and observed continuously. 

Thirty minutes after sedation, post-sedation data (PS) were recorded. The quality of sedation was scored at BL and PS using 3 different scales by evaluation of spontaneous posture, palpebral reflexes, position of the eye globes, jaw tone, response to sound, resistance to physical restraint in lateral recumbency and response to pain [2,19,27]. Scale A [27] is a numerical scale from 0 to 3, with 0 being equivalent to no signs of sedation and 3 representing the highest sedation status. Scale B [2] is a composite numerical scale that assigns a sedation score based on the sum of scores from 4 variables. The scoring system ranges from 0 to 13, with 0 being a bright, alert and responsive dog and 13 indicating the deepest level of sedation. The scoring system of Scale C has been previously described [28] and was adapted from another source [19]. It is a composite numerical scale based on the evaluation of 6 variables and their subscales. The total score ranges from 0 to 18. Scores of 0 to 3 indicate poor sedation, scores of 4 to 6 indicate mild sedation, scores of 7 to 10 indicate moderate sedation, scores of 11 to 15 indicate deep sedation and scores > 15 indicate an anesthesia state. Sedation was judged as poor (1), mild (2), moderate (3), deep (4) or achieving an anesthesia state (5). Adverse events during the procedure, such as tremors, paddling, vocalization, nystagmus, hypersalivation, dysphoria or auditory hyperesthesia (defined as an exaggerated response to a hand clap), were recorded. The Sedation scale A, B and C can be found in the Appendix A [28].

### 2.3. Measured Parameters

The recorded cardiorespiratory parameters included HR, determined by cardiac auscultation; RR, determined by visualization of thoracic movements; and rectal temperature measured using a digital thermometer. In addition, the values of IBP, systolic arterial pressure (SAP), diastolic arterial pressure (DAP) and mean arterial pressure (MAP) were obtained from an arterial catheter that was connected to a pressure transducer. The transducer was previously zeroed, positioned at the xiphoid cartilage level and connected to a multi-parameter monitor (BBraun VetCare SA model 9200, Smith Medical, Waukesha, WI, USA). Arterial blood samples were collected from the pedal arterial catheter using a one-milliliter heparinized syringe. Samples were analyzed using a blood gas analyzer (Idexx VetStat Analyzer, Westbrook, ME, USA) immediately after sampling. The pH, partial pressure of arterial oxygen (PaO_2_), oxygen saturation (StO_2_) and partial pressure of carbon dioxide (PaCO_2_) values were registered after correction for rectal temperature.

The echocardiographic examination was performed using an ultrasound device (Esaote MyLab TM 60 X-Vision, Genoa, Italy) with frequency selection based on the patient’s size. During the acquisition of right parasternal long- and short-axis views, subxiphoid view and left parasternal long view, an electrocardiogram was recorded. M-mode, two-dimensional (2D), and pulsed-wave (PW) Doppler were used for quantitative analysis of the heart. M-mode measurements of the following parameters were registered: the left ventricular diameter in diastole (LVIDd), left ventricular diameter in systole (LVIDs), left ventricular shortening fraction (FS) (FS% = [(LVIDd – LVIDs) / LVIDs] × 100), left atrial diameter in early systole (LA), aortic diameter in diastole (Ao) and left atrial/aortic ratio (LA/Ao). Using the 2D mode, the end-systolic volume (ESV) and end-diastolic volume (EDV) were measured using the modified Simpson method, and the ejection fraction (EF) (EF% = [(EDV – ESV) / EDV] × 100) was calculated. The PW Doppler was used to measure the aortic flow (Ao Vmax), pulmonic flow (PA Vmax), early rapid ventricular filling wave (E-wave), late filling from atrial contraction (A-wave) and E-wave/A-wave ratio (E/A).

Vertebral Heart Score (VHS) determination was performed using the thoracic right lateral view radiograph, acquired at the peak of inspiration. The long and short axes of the heart were transposed onto the vertebral column and recorded as the number of vertebrae, starting at the cranial edge of T4.

A 6-lead ECG was recorded using computerized electrocardiography (Cardioline ar600 view VET, Trento, Italy), and the PR interval and corrected QT interval were determined using a logarithmic formula (QTc = 600 × log QT/RR log).

### 2.4. Statistical Analysis

A required sample size of 20 animals was determined (G*Power, v.3.1.9.2., Dusseldorf, Germany). The sample size was calculated based on the detection of a variation in mean blood pressure of 10 mmHg, assuming a mean value of 90 mmHg and a standard deviation of 15 mmHg, a power of 80% and an alpha error of 0.05.

Statistical analysis was performed using IBM Statistics SPSS v25 (IBM^®^ SPSS^®^ Statistics for Windows, version 25.0, IBM Co., Armonk, NY, USA). The normality of the data distribution was assessed using a Shapiro–Wilk test. A paired-sample t-test was performed to detect differences between baseline and post-sedation values for each registered parameter. Ordinal data were analyzed using the Wilcoxon signed-rank test. Normal variables were reported as mean ± SD, with ordinal variables as median (P25–P75) and nominal variables as percentage. Statistical significance was set at *p* ˂ 0.05.

## 3. Results

### 3.1. Sedation Quality and Side Effects

Sedation quality at baseline was scored as 0 in all dogs. Thirty minutes after the administration of sedation, the median score value for Scale A was 2 (1.5–2), which was equivalent to moderate sedation. For Scale B, the median value was 9 (7.5–11), and for Scale C, the median value was 3 (3–4), equivalent to moderate sedation. For this reason, significant differences in sedation were detected with the A (2: 95% CI 1.5–2; *p* = 0.001), B (9: 95% CI 8–10; *p* = 0.001) and C (3.5: 95% CI 3–3.5; *p* = 0.001) scales. Adverse events were registered after sedation. Tremors were detected in 6/21 (28.6%), nystagmus in 5/21 (23.8%) and auditory hyperesthesia was detected in 4/21 (19%) dogs. 

### 3.2. Cardiorespiratory Parameters

Results of cardiorespiratory parameters are shown in Table 1. The HR, blood pressure (SAP, MAP and DAP), pH, PaO_2_ and StO_2_ parameters decreased significantly after the administration of alfaxalone and butorphanol compared to baseline measurements. The PaCO_2_ increased, compared to baseline values.

### 3.3. Echocardiographic Parameters

All echocardiographic results are displayed in Table 2. A significant decrease from baseline measurements was observed for the LVIDd, Ao, EDV, Ao Vmax, PA Vmax, E-wave and A-wave values. The LA/Ao and E/A ratio showed a significant increase in sedated compared to conscious dogs. No significant differences were observed in the LVIDs, FS, LA, ESV and EF parameters.

### 3.4. X-ray and Electrocardiographic Parameters

The results are displayed in Table 2. The median vertebral heart score did not show a significant difference between conscious and sedated dogs. The electrocardiographic measurements, PQ interval and corrected QT interval increased significantly after sedation. No morphological alterations in the ECG were detected.

## 4. Discussion

The present study evaluates the cardiorespiratory and sedative effects as well as the effects on conventional echocardiographic measurements and cardiac size on radiography and electrocardiography parameters after administration of the intramuscular combination of alfaxalone and butorphanol in healthy dogs. This combination of alfaxalone and butorphanol resulted in moderate sedation 30 min after its intramuscular administration, allowing cardiovascular examination that revealed mild cardiovascular depression, mild echocardiographic measurement alterations and mild ECG changes without alterations in cardiac size on X-ray images.

The quality of sedation in dogs that received intramuscular alfaxalone was dose-dependent [17], with a deep sedation quality obtained 30 min after administration of alfaxalone at 4 mg/kg [23]. However, when a dose of 2 mg/kg was used in dogs, a high variability in sedation scores was described [29]. Deep sedation scores were described in dogs when alfaxalone in a range of 1.5 to 2.5 mg/kg was combined with both butorphanol and sedatives [18,21,22]. In our population, the three scales showed moderate sedation scores with minimal variability in the degree of sedation among individuals. The combination was sufficiently effective to allow for a cardiovascular examination to be performed with minimal restraint.

Adverse effects are commonly observed in dogs when intramuscular alfaxalone is administered. An intramuscular dose of 5 mg/kg produced muscular tremors and a transient staggering gait in all dogs during the recovery period [17]. Similarly, in another study, half of the dogs that received 2 mg/kg of alfaxalone intramuscularly showed side effects [29]. Another study [23] also reported a high frequency (90%) of undesirable effects, such as tremors or paddling. There is evidence that the incidence of adverse effects is lower when alfaxalone is combined with sedatives or opioids. One study [21] reported a better recovery when alfaxalone (2.5 mg/kg) was combined with medetomidine (2.5 μgr/kg) and butorphanol (0.25 mg/kg) compared to alfaxalone alone [17]. When alfaxalone (2 mg/kg) and butorphanol (0.4 mg/kg) were combined with midazolam, dexmedetomidine or acepromazine, a deep degree of sedation was described; however, the combination with midazolam produced a higher frequency of adverse effects [18]. This was also described when alfaxalone and methadone were combined with midazolam [26]. The side effects observed in our study were similar to those reported in other investigations, but their incidence was lower than that described in the literature when alfaxalone alone was used intramuscularly [17,23,29]. Moreover, the incidence of tremors was lower in our dogs compared to the incidence (five out of six beagle dogs) described when a combination of alfaxalone, medetomidine and butorphanol was used in dogs [21]. Moreover, nystagmus was described in two out of six dogs when alfaxalone and butorphanol were combined with midazolam, and this side effect was not described when dexmedetomidine or acepromazine was administered instead of midazolam [18]. The incidence of nystagmus in our population was lower compared to that reported in a previous study [18], which could have been a result of the addition of midazolam.

Thirty minutes after the start of sedation, the cardiorespiratory status was clinically acceptable, even when some of the studied parameters changed significantly. Alfaxalone’s effect on HR was variable. The administration of alfaxalone at 2 mg/kg intramuscularly did not change HR or RR significantly in healthy dogs [29]; moreover, its administration at 2.5 mg/kg and 5 mg/kg resulted in minimal cardiorespiratory effects [17]. However, the intramuscular administration of 4 mg/kg of alfaxalone in healthy dogs produced an increase in HR due to a decrease in blood pressure [23]. In our study population, since a low dose (2 mg/kg) of alfaxalone was used, the reduction in the HR may have been due to the increased vagal tone consequence of the butorphanol administration [30]. An increased vagal tone has been described when alfaxalone was combined with methadone, and it was associated with a decreased HR in healthy beagles [31]. The intravenous administration of a supraclinical dose of alfaxalone (20 mg/kg) was associated with hypotension in dogs [16], but this was not reported when alfaxalone was administered intramuscularly at 10 mg/kg [17] or at 4 mg/kg [23]. Moreover, the combination of alfaxalone with other sedatives [18,22] produced significant alterations in the baseline arterial blood pressure, but hypotension (MAP < 60 mmHg) was not observed. The same was shown in this study since the IBP decreased significantly after sedation, but it remained in the normal clinical range. This may have been associated with a reduction in the peripheral vascular resistance or it may have been due to the sedation reducing the stress associated with manual restraint.

The RR did not change, and despite a significant increase in PaCO_2_ at T30, the value remained within clinically acceptable limits, similar to the results of other studies [21,32]. Patients showed a slight decrease in pH, which could have been secondary to the increased PaCO_2_ and the consequent increase in H+ [33]. Hypoxemia was not observed in our study. Hypoxemia associated with a decrease in RR was reported after the administration of 5, 7.5 and 10 mg/kg of alfaxalone [17] and when alfaxalone was administered in combination with medetomidine (10 μg/kg) and butorphanol (0.1 mg/kg) [22]. In contrast, the administration of a lower dose of alfaxalone [23,29] or the combination of alfaxalone, butorphanol and a low dose of medetomidine [21] did not cause significant changes in oxygenation in dogs. 

Some echocardiographic parameters were slightly changed after the administration of sedation. The significant decrease in EDV and LVIDd suggests a reduction in vascular resistance that would induce a decrease in preload mediated by the drug combination. On the other hand, the reduction in preload could have been due to an increased splenic volume, as a consequence of relaxation of the splenic capsule and subsequent splenomegaly, which has been described in dogs after administration of alfaxalone [34] and in cats after administration of alfaxalone and butorphanol [35]. The decreased preload could explain the reduced Ao Vmax, PA Vmax and the reduced velocity of E-waves and A-waves. In our study, the LV systolic function was conserved after the sedation. However, markers of left ventricular systolic function, FS and EF, were significantly reduced in cats after intramuscular administration of alfaxalone (2 mg/kg) and butorphanol (0.2 mg/kg) [36]. This reduction in LV contractility was not reported when alfaxalone was used alone [23] or in combination with butorphanol and midazolam [32], but the combination of alfaxalone and medetomidine produced a depressant effect in systolic function in dogs [22].

When dexmedetomidine was administered intravenously in healthy dogs [37], an increase in cardiac size, determined with X-ray images, was described. The authors suggested this was due to the vasoconstrictive effects of dexmedetomidine. To the author’s knowledge, there are no studies evaluating the action of other sedatives on cardiac size as determined with X-ray images. In this study, VHS did not change significantly after sedation, likely because the combination of alfaxalone and butorphanol did not produce a vasoconstrictive effect.

The intravenous administration of methadone and hydromorphone caused prolongation of the PR interval and QT interval in healthy dogs [38]. In our study, the PR interval was increased significantly at T30 but remained within normal limits. However, the QTc interval increased significantly and above the normal range, but no malignant arrhythmias were observed during the procedure. The prolongation of the QT interval > 20 ms substantially increases the arrhythmogenic risk in humans [39]; however, the degree of prolongation that can increase the arrhythmogenic risk in dogs has not been defined [38]. The mechanism of QTc interval prolongation in this study is unknown and warrants further investigation.

The present study has several limitations. The study population was calculated based on blood pressure, so the sample size may have limited the power of detecting changes in other tests. In addition, the population studied consisted of healthy Spanish greyhound dogs in an attempt to minimize breed variations. Consequently, the effects of this protocol on patients with heart disease or other breeds are not known, so clinicians should be cautious when administering such a sedative combination to unhealthy dogs, especially those with cardiac disease. Lastly, no control group was used for comparison in this study.

## 5. Conclusions

The intramuscular administration of alfaxalone (2 mg/kg) and butorphanol (0.2 mg/kg) to healthy Spanish greyhound dogs produced adequate sedation to permit cardiovascular examination with mild cardiovascular depression, mild echocardiographic measurement alterations and no alterations in cardiac size on X-ray images. However, previously described side effects, with tremors being the most frequent, were detected. Moreover, this combination must be used with caution in patients with prolongation of the QT interval due to the possible arrhythmogenic potential.

## Figures and Tables

**Table 1 animals-13-02937-t001:** Data of cardiopulmonary parameters at baseline and 30 min after sedation.

Parameter	Baseline	Post-Sedation (T30)	*p*-Value
HR (bpm)	100 ± 27	89 ± 26 *	0.012
RR (bpm)	30 ± 10	27 ± 12	0.397
T (ºC)	38.4 ± 0.6	38.3 ± 0.7	0.327
SAP (mmHg)	169 ± 32	146 ± 21 *	0.001
MAP (mmHg)	99 ± 15	79 ± 10 *	0.001
DAP (mmHg)	71 ± 13	56 ± 13 *	0.001
pH	7.40 ± 0.03	7.37 ± 0.03 *	0.001
PaCO_2_ (mmHg)	34 ± 3.3	40 ± 3.5 *	0.001
PaO_2_ (mmHg)	110 ± 13.2	97 ± 5.8 *	0.001
StO_2_ (%)	97 ± 1.0	96 ± 1.2 *	0.001

HR: heart rate. RR: respiratory rate. T: Temperature. SAP: Systolic arterial pressure. MAP: mean arterial pressure. DAP: diastolic arterial pressure. PaCO_2_: arterial partial pressure of carbon dioxide; PaO_2_: arterial partial pressure of oxygen. StO_2_: oxygen saturation. Data are presented as (Mean ± SD). * Significant differences with respect to baseline data (*p* < 0.05).

**Table 2 animals-13-02937-t002:** Echocardiographic (M-Mode, 2D-Mode and PW-Doppler measurements), X-ray and electrocardiographic measurement data at baseline and 30 min after sedation.

Measurement	Baseline	Post-Sedation (T30)	*p*-Value
IVSd (mm)	12.9 ± 2.1	13.3 ± 2.6	0.641
LVIDd (mm)	44.7 ± 4.5	43.5 ± 3.6 *	0.010
LVWd (mm)	12.2 ± 1.9	12.3 ± 1.6	0.778
IVSs (mm)	18.8 ± 2.5	18.5 ± 2.9	0.880
LVIDs (mm)	27.1 ± 3.8	26.9 ± 4.2	0.601
LVWs (mm)	17.0 ± 2.3	17.3 ± 2.1	0.386
FS (%)	39 ± 6	39 ± 6	0.294
LA (mm)	27.9 ± 3.1	27.8 ± 2.4	0.522
Ao (mm)	23.4 ± 2.7	22.6 ± 2.5 *	0.001
LA/Ao	1.19 ± 0.11	1.23 ± 0.10 *	0.030
EDV (ml)	83 ± 24	77 ± 20 *	0.012
ESV (ml)	29 ± 12	29 ± 11	0.949
EF (%)	65 ± 7	63 ± 8	0.207
Ao Vmax (m/s)	−1.95 ± 0.20	−1.73 ± 0.20 *	0.001
PA Vmax (m/s)	−1.25 ± 0.16	−1.11 ± 0.17 *	0.001
E-wave (m/s)	0.92 ± 1.30	0.81 ± 0.18 *	0.030
A-wave (m/s)	0.58 ± 0.10	0.48 ± 0.09 *	0.010
E/A	1.64 ± 0.40	1.83 ± 0.60 *	0.004
VHS	10.8 ± 0.5	10.8 ± 0.4	0.882
PQ interval (ms)	99 ± 16	110 ± 18 *	0.003
QTc interval (ms)	225 ± 17	244 ± 18 *	0.001

IVSd: interventricular septum in diastole. LVIDd: left ventricular diameter in diastole. LVWd: left ventricular posterior wall in diastole. IVSs: interventricular septum in systole. LVIDs: left ventricular diameter in systole. LVWs: left ventricular posterior wall in systole. FS: left ventricular shortening. LA: left atrial diameter in early systole. Ao: aortic diameter in diastole. LA/Ao: left atrial/aortic ratio. EDV: left ventricular end-diastolic volume. ESV: left ventricular end-systolic volume. EF: ejection fraction. Ao Vmax: peak aortic flow velocity. PA Vmax: peak pulmonic flow velocity. E-wave: peak velocity of early diastolic transmitral flow. A-wave: peak velocity of late diastolic transmitral flow. E/A: E wave/A wave ratio. VHS: Vertebral Heart Score. PQ: PQ interval. QTc: corrected QT interval. Data are presented as (Mean ± SD). * Significant differences with respect to baseline data (*p* < 0.05).

## Data Availability

The full data set can be made available upon request by the corresponding author.

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
