# Peer review of "Sedation Quality and Cardiorespiratory, Echocardiographic, Radiographic and Electrocardiographic Effects of Intramuscular Alfaxalone and Butorphanol in Spanish Greyhound Dogs"

_animals, 2023, doi:10.3390/ani13182937_

Round 1
Reviewer 1 Report
The study is interesting and certainly a contribution to veterinarians in routine clinical practice, therefore I suggest inserting and commenting on the following bibliographic notes even if it concerns the use of alfaxan and butorphanol in other animal species:
DOI 10.5152/actavet.2021.20041
DOI 10.1111/eve.13040
DOI 10.1016/j.vaa.2018.08.006
Interlandi C., Leonardi F., Spadola F., Costa G. L. (2021). Evaluation Of The Paw Withdrawal Latency For The Comparison Between Tramadol And Butorphanol Administered Locally, In The Plantar Surface Of Rat, Preliminary Study. Plos One, P. 2-8, ISSN: 1932-6203, Doi: 10.1371
Leonardi F, Costa G, Stagnoli A, Zubin E, Boschi P, Sabbioni A, Simonazzi B. (2019). The Effect Of Intramuscular Dexmedetomidine-Butorphanol Combination On Tear Production In Dogs. Canadian Veterinary Journal, vol. 60, p. 55-59, ISSN: 0008-5286
Claudia, Interlandi, Gioacchino, Calapai, Bernadette, Nastasi, Carmen, Mannucci, Manuel, Morici, Giovanna L. , Costa (2017). Effects of Atipamezole on the Analgesic Activity of Butorphanol in Rats. JOURNAL OF EXOTIC PET MEDICINE, vol. 26, p. 290-293, ISSN: 1557-5063, doi: DOI: http://dx.doi.org/10.1053/j.jepm.2017.07.001
Line 125 how many observers were assigned the sedation score. Declare the agreement between them W.
Line 186 Describe sedation in more detail by quality and duration. I recommend including a table as well.
Line 185 Declare sample power and data distribution.
Minor editing of English language required
Reviewer 2 Report
Dear Author/s
It has been a pleasure to me reading your manuscript. Although it is a good article, there are few changes that must be corrected for further publication:
Please change "Galgo Español" for "spanish greyhound", or if you prefer in "italics". Check the whole manuscript.
Abstract - Line 41 - change x-rays for radiographic images
Introduction
Lines from 74 to 80 - Lack of literature in the paragraph:
https://doi.org/10.1292/jvms.15-0159
https://doi.org/10.2460/ajvr.81.1.65
https://doi.org/10.1292/jvms.20-0330
https://doi.org/10.1292/jvms.15-0065
Lines 79 to 80 - "has not been studied" which effects has not been studied?in all species?in dogs? are you sure for it?
Statistical analysis - Lines 173 to 176 - why did you calculate with BP values? why ecocardiographic findings were not used? If you use them, how much is your sample size?
Discussion - Line 231 - "the first one to measure cardiorespiratory and sedative effects.." this is not true, there are a couple of previous references (https://doi.org/10.1292/jvms.20-0330; https://doi.org/10.1292/jvms.15-0159)
Conclusions . Line 338 . either you follow "mg kg-1" or "mg/kg", please check author's guidelines for it.
Regards
Overall good quality of English
Reviewer 3 Report
thank you for an interesting paper.
A couple of questions- did any of the side effects require treatment?
Do you recommend giving the drug combination a 30 minutes onset time for the best effects when given IM.
It may be helpful to have your sedation scales as an appendix so we can see the actual parameters measured and their value.
You recommend this drug combination for heart echo with minimal effect on the cardiovascular system?
Round 2
Reviewer 1 Report
Dear author, I saw that you followed my recommendations and your answers are satisfactory.Reviewer 2 Report
Dear Author/s
Thank you for the changes you have done in your manuscript, and I consider it ready for its publication.
Kind regards
Reviewer 3 Report
the authors have addressed my concerns